# *Trans* Fatty Acids Content in Whole-Day Diets Intended for Pregnant and Breastfeeding Women in Gynaecological and Obstetric Wards: Findings from the Study under the “Mum’s Diet” Pilot Program in Poland

**DOI:** 10.3390/nu14163360

**Published:** 2022-08-16

**Authors:** Edyta Jasińska-Melon, Hanna Mojska, Beata Przygoda, Katarzyna Stoś

**Affiliations:** Department of Nutrition and Food Nutritional Value, National Institute of Public Health NIH—National Research Institute, 24 Chocimska Street, 00-791 Warsaw, Poland

**Keywords:** *trans* fatty acids, GC-MS, whole-day hospital diet, pregnant women, breastfeeding women

## Abstract

*Trans* fatty acids (TFAs) have been proven to have an adverse effect on human health by interfering with n-3 long-chain polyunsaturated fatty acids (n-3 LC-PUFA) synthesis. LC-PUFA n-3 are necessary for the development and maturation of the nervous system and retina during the prenatal period and infancy. TFAs are not synthesized de novo in the human body. Their presence in body fluids arises from the diet. The aim of our study was to determine the content of TFAs in individual meals and in a whole-day hospital diet intended for pregnant and breastfeeding women. Samples were collected from six different hospitals in Poland which voluntarily applied to the “Mum’s Diet” Pilot Program. The content of fatty acids, including TFAs, was determined by gas chromatography coupled with mass spectrometry (GC-MS). The TFAs content in the whole-day hospital diets ranged from 3.86 to 8.37% of all fatty acids (% *wt*/*wt*). Food products served for elevenses and afternoon snacks contributed the highest amounts of TFAs. These mainly included dairy products containing TFAs of natural origins. The estimated average intake of TFAs with the hospital diet was 0.72 g/person/day (range: 0.34–1.16 g/person/day) and did not exceed the maximum level of 1% of dietary energy recommended by the World Health Organization.

## 1. Introduction

*Trans* fatty acids (TFAs or ‘trans fats’) are defined as “fatty acids with at least one non-conjugated (namely interrupted by at least one methylene group) carbon-carbon double bond in the trans configuration” [1]. Based on their source in the diet, trans fats are divided into natural (ruminant *trans* fatty acids, r-TFAs) and industrial (industrially produced *trans* fatty acids, i-TFAs). The former are produced naturally in the rumen of ruminant animals through biohydrogenation. Their sources in the human diet include milk and meat from ruminant animals. On the other hand, trans isomers of industrial origin are mainly formed during industrial hydrogenation (‘hardening’) of vegetable oils and during deodorization processes. Their main dietary sources are partially hydrogenated vegetable oils (PHVOs) and food products containing them, such as confectionery and food concentrates. Small amounts of TFAs are also formed during the thermal processing of food, such as frying and baking [2,3]. In food products, trans isomers of oleic acid (18:1) are most abundant, followed by trans isomers of linoleic acid (18:2 n6), α-linolenic acid (18:3 n3), and palmitoleic acid (16:1). It should be noted that the content of TFAs varies significantly both qualitatively and quantitatively between milk fat and partially hydrogenated fats. Among all 18:1 positional trans isomers (Δ4 to Δ16 trans 18:1) identified in food products, vaccenic acid (18:1 Δ11t) is the most abundant in milk fat, accounting for about 55% of all the trans isomers of oleic acid [4]. In partially hydrogenated vegetable fats, the three positional isomers are present in a similar amount of about 20%. These are elaidic acid (18:1 Δ9t), vaccenic acid, and 18:1 Δ10t [2,5,6]. The fatty acids content in food, body fluids, and tissues is most often determined in the form of methyl esters by gas chromatography, usually coupled with a flame ionization detector (GC-FID) or, less often, coupled with a mass detector (GC-MS). It should be noted that so far, no analytical method has been able to distinguish between *trans* fatty acids derived from milk fat and from partially hydrogenated vegetable fats [5].

The content of TFAs in milk and dairy products as well as meat and meat products is relatively low and does not exceed the level of 6% to 10% of all fatty acids. Small variations in the content of TFAs associated with the season and the resulting type of livestock feed used have been observed [5,7]. In contrast, the content of TFAs in products containing partially hydrogenated fats is variable and depends on the quality and quantity of fat used and on the hydrogenation conditions. At the end of the last century, the TFAs content in partially hydrogenated fats reached even more than 50% of total fatty acids [8,9]. Currently, in most products with the addition of partially hydrogenated fats, their content does not exceed 2 g TFAs/100 g fat [10,11,12,13]. It results from the fact that food manufacturers have been reformulating their products as a result of an obligation to state TFAs content on the label, which was introduced in certain countries, such as the United States, Canada, and South American countries [14,15]. On the other hand, the European Union introduced requirements for the maximum permissible content of industrially produced *trans* fatty acids (i-TFAs) at the level of 2 g per 100 g of fat [16].

*Trans* fatty acids have adverse effects on health, including increased LDL (low-density lipoprotein) cholesterol and reduced HDL (high-density lipoprotein) cholesterol level in blood serum, which raises the risk of developing cardiovascular diseases, e.g., coronary heart disease [2]. Numerous studies [17,18,19,20,21,22] have demonstrated a relationship between high dietary intake of TFAs and an increased risk of developing ischaemic heart disease (IHD) and other cardiovascular diseases, as well as obesity [11,13] and type 2 diabetes [23]. Wang et al. [24] estimated that the presence of industrially produced *trans* fats in the diet is the cause of approximately 540,000 deaths per year worldwide due to IHD. On the other hand, De Souza et al. [23] estimated that high TFAs consumption increases the risk of death from all causes by 34%, death from IHD by 28%, and the development of IHD by 21%. In addition, the results of a prospective study, performed on a group of more than 1000 people, indicate a relationship between TFAs intake and an increase in aggressive behaviour in humans [25]. Li et al. [26] also indicated in a recently published paper that the presence of *trans* fatty acids in the diet increases the risk of developing depression in middle-aged women. Numerous studies [27,28,29,30,31] have shown that TFAs interfere with the synthesis of omega-3 and omega-6 long-chain polyunsaturated fatty acids, including docosahexaenoic acid (DHA, C22:6 n-3) and arachidonic acid (AA, C20:4 n-6) of their precursors–α-linolenic acid (18; n-3) and linoleic acid (18:2 n-6), respectively. The TFAs can disrupt the biosynthesis of docosahexaenoic acid, which results in low levels of DHA in the tissues, which in turn can lead to disorders in the development of the nervous system and the retina [32,33]. It seems that TFAs, through their inhibitory effect on LC-PUFA synthesis, may shorten the duration of pregnancy and contribute to the low birth weight of the newborns [27,34,35,36,37]. Furthermore, in infants and young children, TFAs may contribute to the development of asthma, allergic rhinitis, and atopic dermatitis [38]. Taking into account the adverse effects of *trans* fatty acids on the human body, the European Food Safety Authority (EFSA) recommends that TFAs intake *“should be as low as possible in a diet within the context of a nutritionally adequate diet”* [2,39]. In turn, the World Health Organization (WHO) recommends that diets should provide a very low intake of *trans* fatty acids, less than 1% of daily energy intake [40].

It should be noted that unsaturated fatty acids in the *trans* configuration cannot be synthesized de novo in the human body. Their presence in tissues and body fluids, including breast milk, is due to the types of food products consumed. *Trans* fatty acids taken up with the maternal diet can be transferred through the placental barrier and directly into the breast milk. TFAs can also be stored in the body fat depots and then released and transferred into breast milk [41,42,43,44]. In different countries, TFAs content in breast milk ranges widely from less than 0.5% [45,46,47], through approximately 1.5–4.5% [48,49,50,51,52,53,54,55,56,57,58], to more than 11% of all fatty acids (% *wt*/*wt*) [59]. Since the half-life of fatty acids in adipocytes is about 680 days [60,61], the content of TFAs in the diet of women, both before pregnancy as well as during pregnancy, and during breastfeeding, may play a role in the occurrence of developmental disorders in intrauterine life and in infancy.

In 2019, the Pilot Program called “Mum’s Diet” (“Dieta Mamy”) was introduced in Poland by the Ministry of Health. This pilot program was designed to evaluate the standard of pregnant and breastfeeding women’s hospital nutrition. The aim of this pilot program was also to create dietary recommendations for gynaecological and obstetric wards. These recommendations were intended to ensure that the meals for women in the first, second, and third trimesters of pregnancy, and in the postpartum period meet the specific energy, macro-, and micronutrient requirements during these physiological periods. Because these recommendations did not specify *trans* fatty acids consumption, the aim of our study was to determine the content of fatty acids, including *trans* isomers, in the individual meals and the whole-day diets intended for pregnant and breastfeeding women in the obstetric and gynaecological wards. We also wanted to assess whether it is possible to prepare single meals and whole-day diets low in TFAs content and whether these diets meet the recommendations for limiting the amount of TFAs consumed during pregnancy and lactation.

## 2. Materials and Methods

### 2.1. Samples Collection and Preparation

The material for the research was whole-day diets collected in 2020 in six gynaecological and obstetric hospitals from various regions of Poland, which voluntarily participated in the Pilot Program of the Ministry of Health in Poland “Mum’s Diet” (“Dieta Mamy”) and started cooperation with our Laboratory. The participation of hospitals in this pilot program was entirely voluntary, as was the sampling of meals for analytical tests to assess their nutritional and energy value. Eleven hospitals from all over the country declared their willingness to cooperate with our Laboratory and sample on a random day during the program period whole-day meals in order to perform analytical tests. Unfortunately, after the program had started, four of them did not send samples, and one hospital sent only a sample of the dinner, which has not been included in the current assessment. Finally, six hospitals participated in the pilot program and started cooperation with our Laboratory. Two of the hospitals participating in the study were located in cities with a population of more than 1 million (Hospital 3 and Hospital 4), another two in cities with a population of more than 100 to 500 thousand (Hospital 1 and Hospital 5), and the last two in cities with a population of less than 100 thousand (Hospital 2 and Hospital 6).

The whole-day hospital diet consisted of 5 types of meals: breakfast, elevenses, dinner, afternoon snack, and supper. Meals in each of the hospitals participating in the Mum’s Diet program were served at similar times of the day, with intervals between meals ranging from 2 to 4 h. Breakfast was served between 8.00 a.m. to 8.30 a.m., elevenses between 10.00 a.m. and 10.30 a.m., dinner between 1.00 p.m. and 1.30 p.m., afternoon snack between 04.00 p.m. and 04.30 p.m. and supper between 6.00 p.m. and 6.30 p.m.. In each of the six hospitals, 10 parallel individual meals of each type were collected on a single day, which totaled 50 meals collected from one hospital. A total of 300 individual meal samples were taken from all 6 hospitals. The samples of all meals were collected in each hospital on a random day during the program period. Samples of whole-day diets intended for pregnant women were obtained from four hospitals. Among them were samples of the diets intended for women in the first trimester (Hospital 5), second trimester (Hospitals 1 and 6), and third trimester of pregnancy (Hospital 2). Samples from the two remaining hospitals (Hospital 3 and Hospital 4) constituted the so-called breastfeeding diet, intended for women in the postpartum period. All samples of meals were collected by employees of individual hospitals.

The samples of each meal were immediately delivered under refrigerated conditions to the laboratory. In the laboratory, each meal was separately weighed and assessed for compliance between the declared composition provided together with the hospital meals. Subsequently, mean laboratory samples were prepared from every 10 parallel meals from one hospital, including beverages. The preparation of the mean laboratory samples involved mixing all the ingredients and then, depending on the needs, crushing, grinding, and homogenizing all the material. Thus, prepared homogeneous mean laboratory samples, after thorough mixing, were divided into samples intended for individual tests and then frozen and stored at a temperature below −20 °C until analyzed, but no longer than 2 weeks. Before the analysis, the samples were thawed at room temperature and thoroughly mixed. Then 5 g was weighed out from each sample for determination of fat and fatty acids content.

### 2.2. Determination of Fat and Fatty Acids Content, including Trans Isomers

#### 2.2.1. Fat Extraction

Fat content in the samples was determined according to PN ISO 1444:2000 Meat and meat products-determination of free fat content [62] by Soxhlet method using the B-811 extraction apparatus with the B-411 pre-hydrolysis attachment by BÜCHI Labortechnik AG. The method involves the hydrolysis of the sample with 4M hydrochloric acid (Avantor Performance Materials S.A., Poland), the extraction of the residues with petroleum ether (Avantor Performance Materials S.A., Poland), the evaporation of the solvent, drying, and finally weighing of the isolated fat. The results are expressed in grams per 100 g of the meal (g/100 g). The final result of the fat content in each tested meal was the mean of two parallel determinations.

#### 2.2.2. Determination of Fatty Acids by Gas Chromatography-Mass Spectrometry

Fatty acids, including *trans* fatty acids, were analyzed in methyl ester form (FAME) by gas chromatography coupled with a mass detector, using the Hewlett-Packard 6890 gas chromatograph with the 5972A MS detector as described earlier [63,64]. In short, sixty microliters (60 μL) of fat were saponified with 1 mL sodium hydroxide in methanol (0.5 N) (Avantor Performance Materials S.A., Poland) for 15 min at 70 ± 1 °C in an electric multi-block heater and subsequently methylated with 2 mL of 1N hydrochloric acid in methanol (Sigma-Aldrich CHEMIE GmbH, St. Louis, MI, USA) for 15 min at 70 ± 1 °C. After cooling to room temperature, fatty acid methyl esters were extracted with 1 mL of isooctane (2,2,4 trimethylpentane) (Avantor Performance Materials S.A., Poland). One microliter of the sample was injected into the chromatography column. The GC-MS analysis was used with a split injector (ratio 1:100), injector, and detector temperatures were 250 °C, and carrier gas was helium (20 mL/s; a pressure of 43.4 psi). The chromatography oven was programmed to 175 °C for 40 min and, thereafter, increased by 5 °C per min until the temperature reached 220 °C, and was held at this temperature for 16 min. The total time of analysis was 69 min. FAMEs separations were performed on the CP Sil 88 fused silica capillary column (100 m × 0.25 mm i.d., film thickness: 0.20 μm; Agilent J & W GC Columns, USA). Peak identification was verified by comparison with authentic standards (Supelco FAME Mix 37 Component; Sigma-Aldrich, USA) and by mass spectrometry. The obtained results were expressed as a percentage by weight (% *wt*/*wt*) of total fatty acids detected with a chain length between 8 and 24 carbon atoms and in g/100 g of fat. The method was validated and accredited by the Polish Centre of Accreditation (accreditation certificate AB 509). The quality criteria of the method were confirmed by using certified reference material: BCR-162R (Soya-maize oil blend, 5.5 g; Sigma-Aldrich, USA) and satisfactory results of participation in proficiency tests (FAPAS). The fatty acid composition was determined in two parallel fat samples. From each sample, 1 µL was applied twice to the chromatographic column using an autosampler.

### 2.3. Statistical Analysis

Fat content is presented in grams per 100 g of meal (g/100 g), grams per meal (g/meal), and grams per whole-day hospital diet (g/whole-day diet). The content of *trans* fatty acids (TFAs), saturated fatty acids (SFAs), monounsaturated fatty acids (MUFAs), polyunsaturated fatty acids (PUFAs), n-3 long-chain polyunsaturated fatty acids (LC-PUFAs n-3), and n-6 long-chain polyunsaturated fatty acids (LC-PUFAs n-6) is presented as the percentage share of these acids in the sum of all the fatty acids (% *wt*/*wt*). In addition, the results of *trans* fatty acid content are presented in g/100 g meal and in a g/whole-day diet. The content of TFAs, SFAs, MUFAs, and LC-PUFAs is also expressed as a percentage of the total dietary energy (% E). Statistical calculations were performed using the Statistica software, ver. 6.0. (StatSoft Inc., Tulsa, OK, USA).

## 3. Results

### 3.1. Samples Characteristics

Weight (g), energy (kcal), and fat content (g) of individual meals and whole-day hospital diets varied between hospitals (Table 1). The median (Me) weight of the hospital diets was 2866.7 g, ranging from 2181.1 kg (Hospital 1) to 3610.5 kg (Hospital 6). Fat content in whole-day hospital diets ranged from 41.1 g (Hospital 5) to 112.1 g (Hospital 3).

Dinners provided the most fat (Me = 21.5 g), followed by suppers (16.8 g), and breakfasts (14.8 g). The energy value of both the whole-day diet and individual meals also varied between hospitals. The median energy value of the whole-day hospital diet was 2176 kcal, ranging from 1793 kcal (Hospital 5) to 2940 kcal (Hospital 3). Dinners were characterized by the highest energy value (Me = 697 kcal) and afternoon snacks the lowest (Me = 145 kcal).

### 3.2. Fat and Fatty Acid Content of Individual Hospital Meals and Whole-Day Hospital Diet

Table 2 shows the results of the fat content in g/100 g meal and the results of the fatty acids content as the percentage share of individual fatty acids in the sum of all identified fatty acids (% *wt*/*wt*).

Fat content differed significantly both between meals within one hospital and between the same meals in different hospitals. The greatest differences were found for the fat content in elevenses, ranging from below the limit of quantification (LOQ = 0.4 g/100 g) (Hospital 1) to 14.2 g/100 g meal (Hospital 3). Similarly, to the fat content, the fatty acid content also differed both between meals within a single hospital and between the same types of meals in different hospitals. The content of saturated fatty acids (SFAs) ranged from 5.17% *wt*/*wt* (elevenses in Hospital 3) to 74.66% *wt*/*wt* (afternoon snack in Hospital 3). The highest amounts of SFAs were determined in the afternoon snack samples (range: 63.71–74.66% *wt*/*wt*), followed by the breakfast samples (range: 43.91–66.81% *wt*/*wt*). Myristic (C14:0), palmitic (C16:0), and stearic (C18:0) acids dominated in the SFAs group. The content of monounsaturated fatty acids (MUFAs) varied from 21.72% *wt*/*wt* (afternoon snack in Hospital 3) to 70.06% *wt*/*wt* (dinner in Hospital 1). The highest amounts of these fatty acids were found in the dinner samples, ranging from 27.14 to 70.06% *wt*/*wt*, followed by the supper samples (range: 27.57–45.71% *wt*/*wt*). Among MUFAs, oleic acid (18:1 cis) was the most abundant. The content of polyunsaturated fatty acids (PUFAs) in the hospital meals also varied widely, from 2.02% *wt*/*wt* (afternoon snack in Hospital 3) to 39.19% *wt*/*wt* (supper in Hospital 1). The highest amounts of PUFAs were determined in the dinner samples (range: 7.77–20.11% *wt*/*wt*) and supper samples (range: 5.36–39.19% *wt*/*wt*). Among PUFAs, linoleic acid (18:2 n-6) and α-linolenic acid (18:3 n-3) occurred in the highest amounts. The content of 18:2 n-6 ranged from 1.41% *wt*/*wt* (afternoon snack in Hospital 4) to 35.46% *wt*/*wt* (supper in Hospital 1) and was several to a dozen times higher than α-linolenic acid (18:3 n-3) (ranged from 0.25% *wt*/*wt* to 4.37% *wt*/*wt*). Long-chain fatty acids n-3 and n-6 families were present in all the meals analyzed. Their content ranged from 0.30% *wt*/*wt* (breakfast and supper in Hospital 3) to 4.51% *wt*/*wt* (dinner in Hospital 1) for n-3 fatty acids and from 1.55% *wt*/*wt* (afternoon snack in Hospital 2) to 35.49% *wt*/*wt* (supper in Hospital 1) for n-6 fatty acids. The omega-6/omega-3 fatty acids ratio ranged from 16:1 (breakfast and supper in Hospital 3 and breakfast in Hospital 5) to 1:1 (elevenses in Hospital 6).

Trans fatty acids content ranged from 0.08% *wt*/*wt* (dinner in Hospital 1) to 2.35% *wt*/*wt* (afternoon snack in Hospital 4). The highest amounts of TFAs were found in the afternoon snack samples, ranging from 1.56% *wt*/*wt* to 2.35% *wt*/*wt*, followed by the breakfast samples, ranging from 1.22% *wt*/*wt* to 1.89% *wt*/*wt*. Dinner samples, on the other hand, had the lowest TFAs content, ranging from 0.08% *wt*/*wt* to 1.43% *wt*/*wt*. It should be noted that, in most of the dinner samples (more than 80%), the TFAs content was below 1% *wt*/*wt*. The greatest variation in the TFAs content was found in the elevenses and supper samples, ranging from below the limit of quantification (LOQ = 0.01% *wt*/*wt*) to 2.24% *wt*/*wt* and from 0.22% *wt*/*wt* to 1.63% *wt*/*wt*, respectively. Among the TFAs, trans isomers of oleic acid (trans C18:1) predominated, accounting for 62% (Hospital 2) to 68% (Hospital 5) of all the identified trans fatty acids (Figure 1), followed by trans isomers of linoleic acid (trans C18:2 n-6), accounting for from 20% (Hospital 2) to 24% (Hospital 1) of all the trans fatty acids. Trans isomers of palmitoleic acid (trans C16:1) were also identified. They accounted for 9% (Hospital 5) to 18% (Hospital 2) of all the TFAs.

Fat was provided from 20.6% (Hospital 5) to 34.3% (Hospital 3) of energy derived from the whole-day hospital diets. Saturated fatty acids intake ranged from 7.7 E% (Hospital 1) to 17.5 E% (Hospital 3). In turn, the percentage share of the daily energy intake from the remaining fatty acids groups varied from 6.5% (Hospital 5) to 17.3% (Hospital 1) in the case of MUFAs and from 2.2% (Hospital 4) to 7.0% (Hospital 1) in the case of PUFAs. The *trans* fatty acids content in the whole-day hospital diet ranged from 0.34 g (Hospital 1) to 1.16 g (Hospital 6). TFAs provided from 0.2% (Hospital 1) to 0.4% (Hospital 6) of total dietary energy.

### 3.3. Fat and Trans Fatty Acids Content of Individual Meals and Whole-Day Hospital Diet

Table 3 shows the fat content in g/100 g meal and *trans* fatty acids content calculated in g/100 g meal, g/meal, and g/whole-day hospital diet.

*Trans* fatty acids content in whole-day hospital diets ranged from 0.34 g (Hospital 1) to 1.16 g (Hospital 6). The TFAs content differed significantly between meals within one hospital. The greatest differences were found in Hospitals 6 and 3, ranging from 0.09 g/meal to 0.52 g/meal and from 0.001 g/meal to 0.34 g/meal, respectively. The TFAs content also differed between the same types of meals in different hospitals. The greatest differences were found in breakfasts, ranging from 0.17 g/meal (Hospital 5) to 0.52 g/meal (Hospital 6), and in dinners, ranging from 0.03 g/meal (Hospital 1) to 0.34 g/meal (Hospital 3). Breakfasts were characterized by the highest TFAs content (Me = 0.24 g/meal), followed by suppers (Me = 0.23 g/meal). For the remaining meals, i.e., elevenses, dinners, and afternoon snacks, the average TFAs content did not exceed 0.10 g/meal. It was also observed that individual meals served in the hospitals assessed had a relatively low content of TFAs.

Figure 2 shows the percentage share (%) of *trans* fatty acids from different meals in the total TFAs content derived from the whole-day hospital diet. The highest TFAs content at most of the hospitals assessed was recorded for breakfasts—from 26% (Hospital 2) to 62% (Hospital 1). The next ones were suppers (from 11% (Hospital 1) to 40% (Hospital 5)), dinners (from 7% (Hospitals 4 and 6) to 35% (Hospital 3)), and afternoon snacks (from 2% (Hospital 5) to 18% (Hospital 1)). Elevenses contributed the smallest amounts of TFAs to the whole-day diet at the hospitals assessed. The percentage share of these meals in the total dietary TFAs content varied from 0.1% (Hospital 3) to 16% (Hospital 5). In the case of the elevenses sample in Hospital 1, as mentioned before, no determination of the TFAs content was performed because the fat content was too low (<LOQ).

Data on the *trans* fatty acids intake with the whole-day hospital diets were compared with the Food and Agriculture Organization (FAO)/World Health Organization (WHO) recommendation [40]—less than 1 percent of daily energy intake (1 E%). The results obtained are shown in Figure 3. *Trans* fatty acids provided from 0.15 (Hospital 1) to 0.37% (Hospital 6) of energy derived from the whole-day hospital diets. The percentage share of TFAs in total dietary energy did not exceed 1 E% and was in line with the recommendations by WHO.

## 4. Discussion

This study is the first, to our knowledge, to show *trans* fatty acids content in whole-day hospital diets intended for pregnant and breastfeeding women in gynaecological and obstetric wards.

The median TFAs content in all the whole-day diets analyzed was relatively low and amounted to 0.72 g, ranging from 0.34 to 1.16 g. Patients of Hospital 6 were most exposed to the harmful effects of *trans* fatty acids (1.16 g TFAs/whole-day diet), followed by Hospital 3 and 2 with 0.98 g and 0.76 g, respectively. In the remaining hospitals 4 and 5, the estimated TFAs content of the whole-day diets was below 0.70 g and amounted to 0.67 g and 0.62 g, respectively. The lowest content of TFAs was noted for the menu of Hospital 1–0.34 g/whole-day diet. The percentage share of the daily energy intake from TFAs ranged from 0.15 to 0.37%. It was in line with the WHO recommendations for TFAs intake (less than 1% of total energy intake) [40].

Our research confirms a favourable change in the dietary intake of TFAs over the past 15-20 years, both globally and in the European Union (EU) countries. In 2010, the estimated intake of *trans* fatty acids worldwide (266 dietary studies representing 113 countries and 82% of the world population) ranged from 0.2 E% (Barbados) to 6.5 E% (Egypt) [65]. Results published 7 years later (population of 29 countries) were lower than those from 2010 and ranged from 0.3 E% (China) to 4.2 E% (Iran). Only seven countries outside the EU (Brazil, Canada, Costa Rica, Iran, Lebanon, Puerto Rico, and the USA) had an intake of TFAs higher than the WHO recommendation (<1 E%). Furthermore, in 16 out of 21 countries that had data on the source of *trans* fatty acids, r-TFAs intake was higher than i-TFAs intake [3]. It should also be mentioned that so far, no analytical method has been able to differentiate between *trans* fatty acids derived from milk fat and partially hydrogenated vegetable fats. For that reason, the contents of r-TFAs and i-TFAs were estimated based on dietary recall including data on the weight of each product constituting the source of r-TFAs and i-TFAs in the diets. The most recent study published in 2019 (data for 195 countries) showed that TFAs intake in most regions of the world was below 0.8 E%. The highest intake of TFAs was reported for North America, Central America, and Latin America (from 0.8 E% to 1.2 E%) [66]. In our study, we obtained even lower results of TFAs intake with the whole-day hospital diet (0.15–0.37 E%). This confirms the positive trend in reducing dietary TFAs intake. This is most likely due to the introduction of national legislation or voluntary measures of regulation (e.g., product reformulation, changing the recipe, or methods of producing solid fats).

In Poland in 1995, Ziemlański and Budzyńska-Topolowska [67] estimated that the average daily intake of TFAs was 10–14 g/person/day. Data presented by Barylko-Pikielna et al. [68] three years later showed that the average TFAs intake in the average Polish dietary ration was lower and amounted to 3.3 g/person/day. The data were estimated based on the results of the survey of household budgets conducted by Statistics Poland. The results of the TFAs content in oils and spreads, cooking fats, and in products containing these fats, as well as in dairy products and in meat and sausages, were also used in calculations. If it was assumed that all processed fats had the highest analytically determined content of TFAs (the worst-case scenario), this intake would increase to about 6.9 g/person/day. The same authors also estimated the TFAs intake in a population of young women from Warsaw. The data were estimated based on dietary data (a 3-day dietary record) and amounted to about 4.2 g/person/day. In 1999, Daniewski et al. [69], using data on fat intake per capita, estimated that TFAs intake was 2.8 g/person/day. Other studies highlight that the dietary intake of TFAs was estimated to be 14 g/person/day in the mid-1990s and had decreased to 2 g/person/day in 2010. Note that a relatively low level of mean dietary TFAs intake (approx. 1 E%) was recorded in Poland in the years 2009/2010. It amounted to 1.2 E% (1.0–1.3 E%) among men aged under 20 and 1.2 E% (1.0–1.4 E%) among women of the same age. Butter and animal-derived products were mentioned among the main sources of *trans* fatty acids, providing 0.359 g r-TFAs/person/day and 0.496 g r-TFAs/person/day, respectively. Significantly higher TFAs intakes were found for products containing industrially produced *trans* fatty acids (1.5g i-TFAs/person/day) and margarines and other vegetable fats (0.988g i-TFAs/person/day) [70]. According to EFSA, there are insufficient data to confirm differences in the effects on human health between r-TFAs and i-TFAs [37]. Furthermore, the WHO analyses show that reducing the total TFAs of natural and industrial origin, by replacing them with MUFAs and PUFAs, reduces the risk of cardiovascular diseases [71].

In relation to the dietary TFAs intake among pregnant or breastfeeding women, literature provides different data. Innis and King [72] estimated the average daily TFAs intake by breastfeeding women in Canada at the level of 6.9 g/person in 1999, which corresponded to 2.5 E%. Food intake analysis in the same country one year later showed that TFAs intake among pregnant women was lower and amounted to 3.4–3.8 g/person/day (1.3 E%) [73]. *Cohen* et al. [74], in their study from 2011 involving more than 1300 pregnant women in the second trimester, obtained even lower values. The mean (±SD) dietary TFAs intake was 2.35 ± 1.07 g/day. In turn, recently published studies showed that the average daily consumption of TFAs among 33 breastfeeding women was 1.56 ± 0.75 g/day, and it accounted for 0.65% ±0.30% of the total dietary energy. The authors cited noted that the TFAs intake was below the maximum intake recommended by the WHO (1 E%) [40]. In Poland, Mojska et al. [49] estimated that the average daily TFAs intake among breastfeeding women from the Masovian Voivodeship was: 5.70 ± 2.32 g/person in spring and 5.81 ±3.21 g/person in autumn. It corresponded to 1.8 ± 0.8 E% and 2.0 ± 0.7 E%, respectively. Comparing our results to the studies cited above, it should be noted that the current average TFAs intake is lower. It confirms a favourable change in the TFAs intake. This is most likely due to product reformulation, national educational campaigns to raise public awareness of TFAs, and the maximum permissible level of i-TFAs imposed by the European Union.

Among the TFAs identified in our study, *trans* isomers of oleic acid (*trans* C18:1) were predominant (62–68% of all the TFAs identified), followed by *trans* isomers of linoleic acid (*trans* C18:2 n-6)–20–24%. *Trans* isomers of palmitoleic acid (*trans* C16:1) were present in smaller amounts (9-18%). Their content in terms of g/day varied widely: for *trans* C18:1–from 0.21 g (Hospital 1) to 0.74 g (Hospital 6), for *trans* C18:2 n-6–from 0.08 g (Hospital 1) to 0.24 g (Hospital 6), and for *trans* C16:1–from 0.05 g (Hospital 1) to 0.17 g (Hospital 6). We compared the data obtained in this study with the results of our own studies from the years 1999–2000 [49] carried out among 100 breastfeeding women from the Masovian Voivodeship. It should be noted that *trans* C18:1 was the dominant *trans* isomer in the diets analyzed in the previous studies as well. The mean (±SD) intake of this acid was 4.33 ± 2.43 g/day in spring and 5.50 ± 2.30 g/day in autumn. According to the authors cited, the main sources of TFAs in the mothers’ diets were bakery products, sweets, and snacks. Additionally, Cohen et al. [74] noted high intakes of C18:1t and C18:2tc isomers (1.78 g/day and 0.33 g/day, respectively) in the pregnant women’s diets. Smaller intakes of C18:2tt (0.13 g/day), C18:2ct (0.12 g/day), and C16:1t (0.11 g/day) have also been observed. The main sources of C16:1t were dairy products and meat products, especially beef, while the sources of C18:2tc were bread and fried foods. On the other hand, according to Daud et al. [55], the most common *trans* isomers consumed in the breastfeeding women’s diet from Malaysia were i-TFAs, i.e., linoelaidic acid (C18:2t9,12)—0.07 g/100 g of food, followed by elaidic acid (C18:1t9)—0.03 g/100 g of food. Based on the studies cited above, it should be concluded that not only the dietary intake of TFAs but also the intake profile of *trans* fatty acids has decreased in recent years. Currently, the main sources of TFAs in the diet are products of natural origin. At the same time, it is worth noting that the whole-day hospital diets analyzed in our study had a low content (less than 1 g) of the sum of *trans* isomers of C18:2 and C16:1. This is important for the normal development of infants and young children. Cohen et al. [74] showed that a combined intake of C16:1t derived from ruminant fat and C18:2tc derived from partially hydrogenated fats of 2.3 g/day during the second trimester of pregnancy was associated with increased fetal growth and, consequently, a higher risk of overweight and diabetes in adulthood.

Typical products that constitute the main source of i-TFAs, namely confectionery and pastry products, hard (cube) margarines, and fast-food products, especially French fries and salty snacks, were not present in the currently studied whole-day hospital diets. In addition, frying, which can lead to the formation of TFAs, was excluded from the thermal preparation techniques used. Literature data suggest that the type of thermal treatment used to prepare meals has an impact on the TFAs intake with the diet. *Trans* fatty acids can form in the cooking processes, primarily frying and baking [37]. We found that the main sources of TFAs among patients of the hospitals evaluated were natural products, i.e., milk and dairy products as well as meat and meat products. Those products are sources of naturally occurring *trans* fatty acids (r-TFAs). In Hospital 1, cottage cheese (breakfast) and natural yoghurt (afternoon snack) occurred once. As far as meat and meat products are concerned, pork loin was served once (dinner). In addition, a frequent supply of fats was observed (butter at breakfast and supper). On the other hand, in Hospital 6 milk and dairy products were served frequently: natural yoghurt twice (elevenses and an afternoon snack), milk once, but in two forms (breakfast: milk soup and grain coffee with milk), and cottage cheese (supper). With regard to meat and meat products, the hospital diet included a double supply of products from this group (breakfast, dinner). In addition, butter was frequently served (breakfast, supper). Taking the above into account, the whole-day diet in Hospital 1 included a considerably lower number of products that are natural sources of TFAs compared to the menu of Hospital 6. Kresic et al. in Croatia [75] also found that, alongside sweets and bread, milk and milk products were the main sources of TFAs in breastfeeding women’s diet. Similar results were noted by Aumeistere et al. [57], who studied the breast milk composition of Latvian women. The authors cited noted an association between the total TFAs content in breast milk and the breastfeeding mothers’ dietary intake of meat and meat products (ρ = 0.296, *p* = 0.021) as well as milk and milk products (ρ = 0.566, *p* < 0.0005). Moreover, Zupanič et al. [76] in their study carried out among 1248 participants from three age groups (10–17, 18–64, 65–74) showed that the main sources of TFAs in the diet, regardless of the age group, were butter, bread, meat dishes, and cold cuts. According to these authors, after the reformulation of products to reduce the i-TFAs content, the main sources of *trans* fatty acids in the diet of the Slovenian population are now foods that are sources of r-TFAs.

Our findings show that, in Poland, there has been a beneficial reduction in the dietary intake of TFAs. These may be due to the product reformulation. Many manufacturers are gradually eliminating vegetable oils subject to catalytic hydrogenation from the production process. In addition, for many years, educational activities have been carried out. Their main goals are to improve the awareness and knowledge of the population in the field of food and nutrition. This is done by providing consumers with appropriate information on products and their composition. In Poland, an example of an extensive campaign to reduce the level of TFAs in food and diet is the National Health Program that was implemented in 2017. This program includes regularly updating the *Database of trans fatty acids content in Polish food* - e-Base of TFAs [10]. 

In this study, we also noted that the percentage share of total dietary energy from fat met Polish recommendations for pregnant and postpartum women (reference 20–35 E%) [77]. Compared to our previous studies [49,78], the fat intake was lower. This may indicate a change in dietary habits over the last 20 years as a result of, among other things, the implementation of nutrition education programs such as the National Health Program [79,80] or the “Mum’s Diet” Pilot Program (“Dieta Mamy”) [81]. Monounsaturated fatty acids were consumed in the amount recommended by FAO/WHO [40], representing a mean of ~11 E%. The high intake of MUFAs in the present study is due to the high content of oleic acid (C18:1) in hospital diets. It should be noted that C18:1 is the primary MUFA in hospital diets as it is present in virtually all plant and animal products [37].

On the other hand, a high proportion of saturated fatty acids was found in the hospital diets studied. In the case of five evaluated whole-day hospital diets (diets in Hospitals 2, 3, 4, 5, and 6), the intake of SFAs exceeded the reference intake values (no more than 10% of the dietary energy, 10 E%) [40,71,77]. The high level of SFAs intake in our study is probably due to far too frequent consumption of butter (2 or 3 times a day) as the main fat used as a bread spread. This product, according to the Polish recommendations [77,79], should be limited. High SFAs intake may also arise from excessive consumption of milk and dairy products. These products should be consumed in the amount of, for example, two glasses of milk per day in order to cover the necessary dose of calcium. However, skimmed and semi-skimmed products should be chosen. Additionally, this study shows that polyunsaturated fatty acids intake in five hospitals was not consistent with the WHO/FAO reference intake values for adults (6–11 E%) [40]. In Hospitals 2, 3, 4, 5, and 6, PUFAs were consumed in a lower amount (range: 2.2–3.7 E%) than recommended. Only in one hospital (diet in Hospital 1) did the intake of these acids (7.0 E%) fall within the above-mentioned range. Furthermore, the ratio of omega-6 to omega-3 fatty acids deviated from the beneficial value (about 2:1). Due to the normal fetal/infant development during pregnancy and lactation, the intake of SFAs should be reduced, while the intake of PUFAs should be increased. As already mentioned, our study reported too high mean saturated fatty acids intake (12.9 E%) and too low mean polyunsaturated fatty acids intake (3.6 E%). This indicates the need for further education on correct dietary behaviours.

Summing up, it should be emphasized that, in accordance with the recommendations of the “Mum’s Diet” Pilot Program, pregnant and breastfeeding women’s diets should include a variety of ingredients and should not be monotonous and uniform. Therefore, the composition of meals was probably changed from day to day. It should be noted that meal samples from each hospital were collected in only one randomly selected day of the program. Therefore, the presented results are to be considered as pilot and research should be continued.

## 5. Conclusions

In all analyzed samples of meals, except for one, *trans* fatty acid content was above the limit of quantification. Analytically determined TFAs content in samples of meals showed that whole-day hospital diets of pregnant and breastfeeding women in gynaecological and obstetric wards were characterized by low content of trans fatty acids. The average TFAs content in whole-day diets was lower than the WHO recommendations. Analysis of meal composition and the analytically determined *trans* fatty acid content showed that the main source of TFAs was milk and dairy products as well as in meat and meat products. To estimate TFAs content in Polish food rations, the *Database of TFAs content in Polish food* [10] may be helpful. The percentage share (%) of the daily energy intake from fat met national recommendations for pregnant and breastfeeding women. However, the hospital diets provided too many SFAs, compared to the recommended standards, and too few PUFAs. Our results related to fat and fatty acids content, including TFAs, indicate that further nutritional education is required. In addition, it should be noted that meal samples analyzed in our study were collected from six hospitals that participated voluntarily in the pilot program. Due to the small number of hospitals participating in the program, further research in this area is required.

## Figures and Tables

**Figure 1 nutrients-14-03360-f001:**
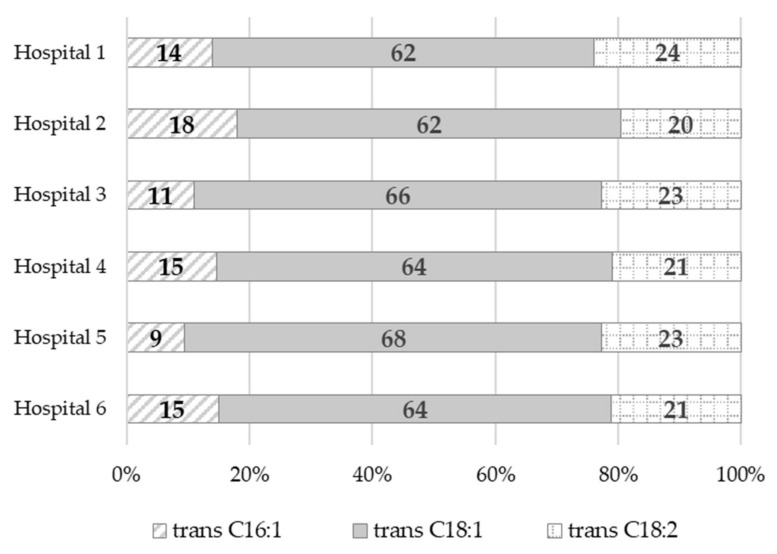
The percentage share (%) of individual *trans* isomers in the group of all the *trans* fatty acids identified in the whole-day hospitals’ diets.

**Figure 2 nutrients-14-03360-f002:**
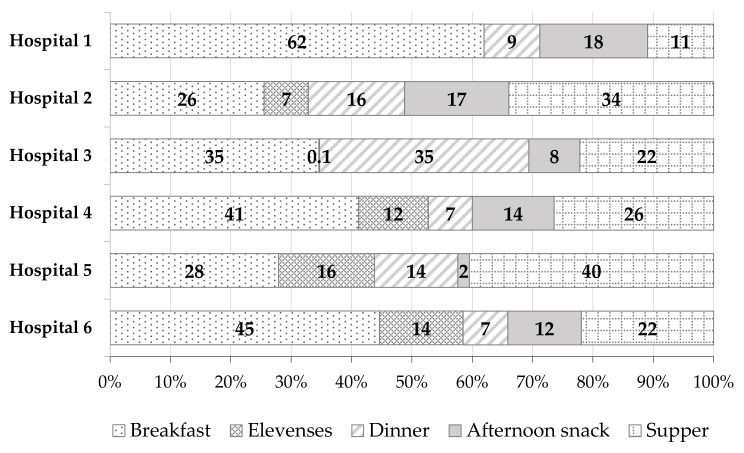
The percentage share (%) of *trans* fatty acids (TFAs) from different meals in the total TFAs content derived from the whole-day hospital diet.

**Figure 3 nutrients-14-03360-f003:**
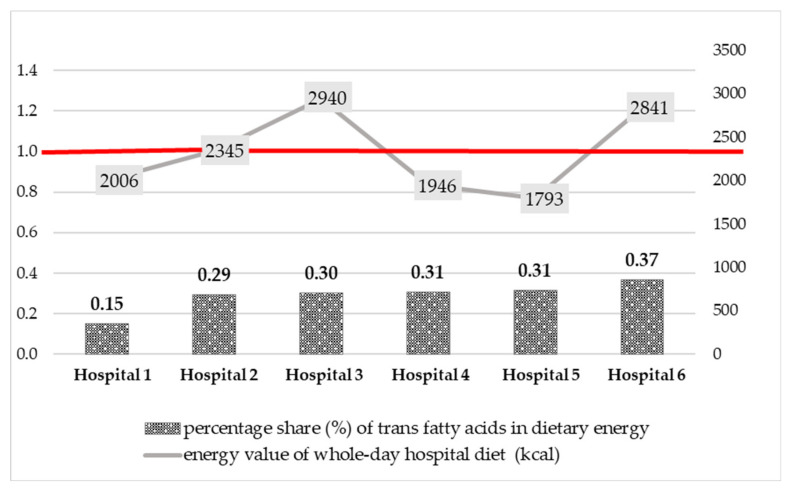
Comparison of the percentage share (%) of *trans* fatty acids in energy derived from the whole-day hospital diets with the WHO recommendation (1% of dietary energy, 1 E%).

**Table 1 nutrients-14-03360-t001:** Comparison of the selected parameters of the individual meals and whole-day hospital diets.

	Hospital 1 ^a^	Hospital 2 ^a^	Hospital 3 ^a^	Hospital 4 ^a^	Hospital 5 ^a^	Hospital 6 ^a^	Average Value (Me, Median)
**Breakfast**							
Weight (g/meal)	538.8	751.4	829.5	906.5	476.0	803.3	**777.4**
Energy value (kcal/meal)	481	687	672	592	646	801	**659**
Fat (g/meal)	12.4	15.0	23.2	14.5	14.3	29.7	**14.8**
**Elevenses**							
Weight (g/meal)	200.0	96.6	128.9	62.5	315.9	447.2	**164.5**
Energy value (kcal/meal)	77	233	382	163	95	253	**198**
Fat (g/meal)	0.4	4.1	18.3	5.1	4.7	7.2	**4.9**
**Dinner**							
Weight (g/meal)	918.8	1184.6	1201.3	1283.4	1005.5	1261.4	**1193.0**
Energy value (kcal/meal)	745	483	1256	613	334	796	**679**
Fat (g/meal)	38.6	17.8	49.3	11.6	6.0	25.2	**21.5**
**Afternoon snack**							
Weight (g/meal)	245.0	282.3	102.5	153.9	150.8	383.1	**199.5**
Energy value (kcal/meal)	127	242	163	79	66	241	**145**
Fat (g/meal)	2.9	6.2	5.3	3.8	0.6	6.5	**4.6**
**Supper**							
Weight (g/meal)	278.5	694.3	514.8	550.0	267.0	715.5	**532.4**
Energy value (kcal/meal)	576	700	467	499	652	750	**614**
Fat (g/meal)	17.5	20.8	16.0	14.3	15.5	27.9	**16.8**
**Weight of whole-day diet** **(g/whole-day diet)**	**2181.1**	**3009.2**	**2777.0**	**2956.3**	**2215.2**	**3610.5**	**2866.7**
**Energy value of whole-day diet (kcal/whole-day diet)**	**2006**	**2345**	**2940**	**1946**	**1793**	**2841**	**2176**
**Fat content in whole-day diet** **(g/whole-day diet)**	**71.8**	**63.9**	**112.1**	**49.3**	**41.1**	**96.5**	**67.9**

^a^ Numbers from 1 to 6 have been assigned to hospitals participating voluntarily in the “Mum’s Diet” Pilot Program.

**Table 2 nutrients-14-03360-t002:** Comparison of fat ^1^ and fatty acid ^2^ content in individual meals of the hospital diets.

Meal Type	Fat [g/100 g] ^1^	Saturated Fatty Acids (SFAs) [% *wt*/*wt*] ^2^				Monounsaturated Fatty Acids (MUFAs) [% *wt*/*wt*] ^2^			Polyunsaturated Fatty Acids (PUFAs) [% *wt*/*wt*] ^2^							*Trans* Fatty Acids (TFAs) [% *wt*/*wt*] ^2^
		C14:0	C16:0	C18:0	Total SFAs	C16:1 *cis*	C18:1 *cis*	Total MUFAs	C18:3 n-3	C22:6 n-3	Total LC-PUFAs n-3	C18:2 n-6 *linoleic acid, LA*	C20-4 n-6	Total LC-PUFAs n-6	Total PUFAs	Total TFAs
**Hospital 1 ^a^**																
Breakfast	2.3	9.86	44.91	7.44	66.79	1.31	22.23	24.57	0.71	0.04	0.86	6.11	<LOQ ***	6.11	6.97	1.67
Elevenses	<LOQ *	- **	- **	- **	- **	- **	- **	- **	- **	- **	- **	- **	- **	- **	- **	- **
Dinner	4.2	0.16	6.26	2.41	9.73	0.27	68.79	70.06	4.37	0.01	4.51	15.36	0.07	15.46	20.11	0.08
Afternoon snack	1.2	10.83	45.87	7.98	70.72	1.39	22.08	24.69	0.32	0.03	0.50	2.07	<LOQ ***	2.13	2.70	1.89
Supper	6.3	1.18	10.85	2.94	16.74	0.40	42.75	43.81	3.37	0.02	3.49	35.46	<LOQ ***	35.49	39.19	0.22
**Hospital 2 ^a^**																
Breakfast	2.0	6.58	34.92	7.83	54.19	1.35	26.64	29.10	1.24	0.04	1.35	13.51	<LOQ ***	13.54	15.33	1.30
Elevenses	4.2	5.90	39.41	8.29	57.47	1.63	27.04	29.74	0.72	0.06	0.98	10.26	<LOQ ***	10.32	11.41	1.38
Dinner	1.5	2.69	32.38	9.57	46.85	1.89	40.26	43.07	0.93	<LOQ ***	1.39	7.76	<LOQ ***	7.87	9.42	0.66
Afternoon snack	2.2	10.93	41.91	10.40	70.56	1.33	22.34	25.00	0.48	0.05	0.67	1.50	<LOQ ***	1.55	2.30	2.13
Supper	3.0	5.00	35.02	8.56	52.20	1.92	31.97	34.80	1.12	0.17	1.88	9.62	<LOQ ***	9.68	11.74	1.26
**Hospital 3 ^a^**																
Breakfast	2.8	8.23	47.17	7.25	66.81	1.33	24.04	26.18	0.26	0.04	0.30	4.82	<LOQ ***	4.82	5.47	1.46
Elevenses	14.2	0.03	3.61	0.86	5.17	0.08	61.55	62.35	3.20	<LOQ ***	3.20	29.23	<LOQ ***	29.23	32.48	<LOQ ***
Dinner	4.1	3.47	43.80	4.10	53.79	0.82	36.27	37.76	0.80	0.03	0.82	6.92	<LOQ ***	6.95	7.77	0.69
Afternoon snack	5.2	11.46	50.92	7.10	74.66	1.11	19.66	21.72	0.25	0.03	0.31	1.68	<LOQ ***	1.69	2.02	1.56
Supper	3.1	6.34	49.20	6.75	65.72	1.51	25.37	27.57	0.26	<LOQ ***	0.30	4.77	<LOQ ***	4.79	5.36	1.36
**Hospital 4 ^a^**																
Breakfast	1.6	10.16	38.41	9.25	65.31	1.45	22.83	25.78	0.86	0.08	0.98	5.68	0.12	5.87	7.02	1.89
Elevenses	8.2	7.69	39.61	8.47	61.00	1.89	27.18	30.28	0.49	<LOQ ***	0.57	6.23	<LOQ ***	6.28	7.20	1.50
Dinner	0.9	1.71	29.60	5.84	40.70	2.30	39.89	43.30	0.98	0.03	1.36	12.85	0.12	13.11	15.48	0.42
Afternoon snack	2.5	11.91	40.77	10.33	71.77	1.35	20.88	23.76	0.37	0.05	0.45	1.41	0.12	1.61	2.10	2.35
Supper	2.6	5.86	33.41	8.02	51.84	1.92	33.79	37.05	0.85	0.02	1.02	7.95	0.24	8.25	9.85	1.23
**Hospital 5 ^a^**																
Breakfast	3.0	4.68	28.25	7.22	43.91	1.11	34.25	36.37	0.97	0.03	1.06	16.87	0.12	16.99	18.50	1.22
Elevenses	1.5	10.04	42.05	8.85	67.38	1.34	24.35	27.06	0.48	<LOQ ***	0.63	2.78	<LOQ ***	2.83	3.46	2.09
Dinner	0.6	5.27	39.38	9.36	58.64	2.13	23.84	27.14	2.17	0.08	2.44	9.32	0.51	10.08	12.76	1.43
Afternoon snack	0.4	8.07	37.69	10.04	63.71	1.21	23.74	26.38	2.33	0.09	2.79	4.77	<LOQ ***	4.90	7.90	2.00
Supper	5.8	9.09	40.47	7.26	61.40	1.62	27.72	30.47	0.77	0.03	0.89	5.48	0.07	5.59	6.48	1.63
**Hospital 6 ^a^**																
Breakfast	3.7	9.11	38.95	9.60	63.88	1.74	24.81	27.85	0.56	0.04	0.83	5.43	<LOQ ***	5.48	6.68	1.74
Elevenses	1.6	11.54	37.77	9.29	66.87	1.46	21.30	24.82	3.14	0.05	3.20	2.74	<LOQ ***	2.83	6.06	2.24
Dinner	2.0	2.35	20.65	4.93	30.79	2.16	46.34	49.64	2.56	0.06	2.63	15.63	0.37	16.09	19.23	0.34
Afternoon snack	1.7	10.25	38.49	9.19	65.60	1.27	21.38	24.33	2.05	0.06	2.17	5.43	<LOQ ***	5.64	7.89	2.17
Supper	3.9	5.02	24.83	6.39	40.48	1.15	43.15	45.71	2.71	0.04	2.82	9.96	<LOQ ***	10.02	12.89	0.91

^1^ Total fat concentration is presented as grams per 100 g [g/100 g]; ^2^ Data for fatty acids are presented as the relative proportion of each fatty acid (% of total fatty acids – % *wt*/*wt*); ^a^ Numbers from 1 to 6 have been assigned to hospitals participating voluntarily in the “Mum’s Diet” Pilot Program, Abbreviation: Total SFAs—sum of saturated fatty acids (C4:0 + C6:0 + C8:0 + C10:0 + C11:0 + C12:0 + C14:0 + 15:0 + C16:0 + C17:0 + C18:0 + C20:0 + C22:0 + C24:0 + C21:0 + C23:0), Total MUFAs—sum of monounsaturated fatty acids (C14:1 + C15:1 + C16:1 cis + C17:1 + C18:1 cis + C20:1 + C22:1 + C24:1), Total LC-PUFAs n-3—sum of long chain n-3 polyunsaturated fatty acids (C18:3 n-3 + C20:3 n-3 + C20:5 n-3 + C22:5 n-3 + C22:6 n-3), Total LC-PUFAs n-6—sum of long chain n-6 polyunsaturated fatty acids (C18:2 n-6 + C18:3 n-6 + C20:3 n-6 + C20:4 n-6), Total PUFAs—sum of polyunsaturated fatty acids (C18:2 n-6 + C18:3 n-6 + C20:3 n-6 + C20:4 n-6 + C18:3 n-3 + C20:3 n-3 + C20:5 n-3 + C22:5 n-3 + C22:6 n-3 + C20:2 + 22:2), Total TFAs—sum of *trans* fatty acids (*trans* C16:1 + *trans* C18:1 + *trans* C18:2), LA—linoleic acid, ALA—α-linolenic acid. * LOQ (limit of quantification) for fat = 0.4 g/100 g—½ LOQ was taken for the calculations, i.e., 0.2 g/100 g. ** the fatty acid content was not determined because the fat content was too low. *** LOQ (limit of quantification) for fatty acids = 0.01% *wt*/*wt*—½ LOQ was taken for the calculations, i.e., 0.005% *wt*/*wt*.

**Table 3 nutrients-14-03360-t003:** Comparison of fat and *trans* fatty acid content in hospital diets in g/100 g, g/meal, and g/whole-day diet.

	Meal Type	Total Fat (g/100 g)	TFAs (g/100 g)	TFAs (g/Meal)	TFAs (g/Whole-Day Diet)
**Hospital 1** ^a^	Breakfast	2.3	0.04	0.21	0.34
	Elevenses	<LOQ *	ND **	ND **	
	Dinner	4.2	0.003	0.03	
	Afternoon snack	1.2	0.02	0.06	
	Supper	6.3	0.01	0.04	
**Hospital 2** ^a^	Breakfast	2.0	0.03	0.20	0.77
	Elevenses	4.2	0.06	0.06	
	Dinner	1.5	0.01	0.12	
	Afternoon snack	2.2	0.05	0.13	
	Supper	3.0	0.04	0.26	
**Hospital 3** ^a^	Breakfast	2.8	0.04	0.34	0.98
	Elevenses	14.2	0.001 ***	0.001 ***	
	Dinner	4.1	0.03	0.34	
	Afternoon snack	5.2	0.08	0.08	
	Supper	3.1	0.04	0.22	
**Hospital 4** ^a^	Breakfast	1.6	0.03	0.27	0.67
	Elevenses	8.2	0.12	0.08	
	Dinner	0.9	0.004	0.05	
	Afternoon snack	2.5	0.06	0.09	
	Supper	2.6	0.03	0.18	
**Hospital 5** ^a^	Breakfast	3.0	0.04	0.17	0.62
	Elevenses	1.5	0.03	0.10	
	Dinner	0.6	0.01	0.09	
	Afternoon snack	0.4	0.01	0.01	
	Supper	5.8	0.09	0.25	
**Hospital 6** ^a^	Breakfast	3.7	0.06	0.52	1.16
	Elevenses	1.6	0.04	0.16	
	Dinner	2.0	0.01	0.09	
	Afternoon snack	1.7	0.04	0.14	
	Supper	3.9	0.04	0.25	

^a^ Numbers from 1 to 6 have been assigned to hospitals participating voluntarily in the “Mum’s Diet” Pilot Program. <LOQ * = LOQ for fat = 0.4 g/100 g. ND **—not determined. *** TFAs content < LOQ = 0.01% *wt*/*wt*—½ LOQ was taken for the calculations, i.e., 0.005% *wt*/*wt*. Abbreviation: LOQ—Limit of Quantification; TFAs—*Trans Fatty* Acids.

## Data Availability

Not applicable.

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
