# Peer review of "Trans Fatty Acids Content in Whole-Day Diets Intended for Pregnant and Breastfeeding Women in Gynaecological and Obstetric Wards: Findings from the Study under the “Mum’s Diet” Pilot Program in Poland"

_nutrients, 2022, doi:10.3390/nu14163360_

Round 1

Reviewer 1 Report

This manuscript report very well the findings of the Polish pilot program, which aims at enhancing healthy diet for pregnant and breastfeeding women.

Some minor remarks:

Please remove the text in parentheses from line 28, as detailed explanation of the structure of the compounds is not necessary in the Introduction.

Please describe the timings of the meals in the Materials and Methods section (lines 137-138), as the timing for dinner may differ.

The meal samples were collected from six hospitals that participated voluntarily in the Mum's Diet program. Were there ay differences how the hospitals were adhering to the diet recommendations?

In Results section, it would be interesting to see a comparison between analysis results and fatty acid content calculated based on the ingredients of the meal.

Please also discuss the fact that meals were sampled only on a single day per each hospital, and that the composition of the meals may vary day-by-day.

Author Response

Replay to the Reviewer #1

Reviewer 2 Report

Dear Authors,

I advise in the text to use the abbreviations (MUFA, TFA, etc.) instead of full expressions, each time when possible, and not only occasionally. 

I am not sure, that in the 11th row the "among other" expression is necessary.

In row 182 please correct the grammar error.

In row 300 and 538 the text color should be black instead of partly red.

According to my opinion the text in row 540-542 is self-evident and overcomplicated. Please rewrite this few sentences.

Author Response

Replay to the Reviewer # 2

This manuscript is a resubmission of an earlier submission. The following is a list of the peer review reports and author responses from that submission.